# Involvement of Road Users from the Productive Age Group in Traffic Crashes in Saudi Arabia: An Investigative Study Using Statistical and Machine Learning Techniques

Md. Kamrul Islam [1,*], Uneb Gazder [2,*], Rocksana Akter [3] and Md. Arifuzzaman [1]

1   Department of Civil and Environmental Engineering, College of Engineering, King Faisal University, Al-Ahsa 31982, Saudi Arabia; marifuzzaman@kfu.edu.sa
2   Department of Civil Engineering, University of Bahrain, Isa Town P.O. Box 32038, Bahrain
3   Department of Civil Engineering, Dhaka University of Engineering and Technology, Gazipur 1707, Bangladesh; rocksananur@duet.ac.bd
*   Correspondence: maislam@kfu.edu.sa (M.K.I.); ugazder@uob.edu.bh (U.G.)

**Abstract:** Road traffic crashes (RTCs) are a major problem for authorities and governments worldwide. They incur losses of property, human lives, and productivity. The involvement of teenage drivers and road users is alarmingly prevalent in RTCs since traffic injuries unduly impact the working-age group (15–44 years). Therefore, research on young people's engagement in RTCs is vital due to its relevance and widespread frequency. Thus, this study focused on evaluating the factors that influence the frequency and severity of RTCs involving adolescent road users aged 15 to 44 in fatal and significant injury RTCs in Al-Ahsa, Saudi Arabia. In this study, firstly, descriptive analyses were performed to justify the target age group analysis. Then, prediction models employing logistic regression and CART were created to study the RTC characteristics impacting the target age group participation in RTCs. The most commonly observed types of crashes are vehicle collisions, followed by multiple-vehicle and pedestrian crashes. Despite its low frequency, the study area has a high severity index for RTCs, where 73% of severe RTCs include individuals aged 15 to 44. Crash events with a large number of injured victims and fatalities are more likely to involve people in the target age range, according to logistic regression and CART models. The CART model also suggests that vehicle overturn RTCs involving victims in the target age range are more likely to occur as a result of driver distraction, speeding, not giving way, or rapid turning. As compared with the logistic regression model, the CART model was more convenient and accurate for understanding the trends and predicting the involvement probability of the target age group in RTCs; however, this model requires a higher processing time for its development.

**Keywords:** road traffic crashes; CART models; age group

## 1. Introduction

Road traffic crashes (RTCs) are a source of concern for authorities and governments all over the world. This is because they contribute to losses, both in terms of property and in terms of human lives and productivity [1]. They were reported to be the eighth leading cause of death for all ages in the last decade [2]. In addition, 20 to 50 million people are injured in RTCs every year [3]. This trend has been increasing in recent years, especially in developing countries, and is predicted to continue to increase if appropriate measures are not taken [4]. RTCs are the leading cause of death for young people, accounting for 1.24 million deaths each year [5,6]. Approximately 85 percent of these annual deaths occur in developing countries. The data led us to the conclusion that there is one RTC-related death every 4 min and one RTC every minute. Countries spend 1 to 2% of their total national product on dealing with RTCs [5]. Despite the fact that developing countries account for only 52 percent of all vehicles on the road, they account for 80 percent of all

road traffic deaths [7]. Pedestrians, passengers, and cyclists account for nearly 90% of road traffic fatalities in developing nations, where most injuries occur in metropolitan areas, and pedestrians in cities account for 55–70 percent of deaths [8,9]. Pedestrian injury rates have been linked to gender, age, population, density, current demographic composition, unemployment, traffic flow, education, and other characteristics, according to a study conducted by La Scala et al. [10]. Mungnimit and Bener [11,12] concluded that reckless driving was the top cause of traffic crashes in Thailand and Qatar in comparable research on the sequence of road traffic incidents and their causes (71%). The bulk of crash casualties (53%) were from society's most productive stratum, those aged 10 to 40. Furthermore, age was a significant influence in all vehicle deaths, accounting for around half of all fatalities [13]. According to the World Health Organization's Global Status Report on Road Safety 2018, issued in December 2018, the number of yearly road traffic deaths has increased to 1.35 million, and road traffic injuries have become the leading cause of death among those aged 5 to 29 years [14].

One of the alarming commonalities in RTCs is the high involvement of young drivers and road users in different countries [15,16]. In developing countries, road traffic injuries disproportionately affect the productive (working) age group (15–44 years) and children. In 1998, the productive age group accounted for 51% of fatalities and 59% of disability-adjusted life years due to road traffic injuries worldwide [17].

Workers below the age of 30 are considered to be healthier and more mobile than those above 30. Workers above the age of 30 become leaders of their families and attain key positions in their organizations. Moreover, their productivity increases till the age of 50 [18]. When the age group and RTC data have been compared, it was found that 55 percent of road traffic crash victims were between the ages of 25 and 65, while the remaining 45 percent of RTC victims were between the ages of 16 and 24 [7]. Moreover, young adults and children are the ones on which the future of a country depends. Evidence has shown that age is a significant factor affecting the involvement of road users in RTCs, with a higher proportion of younger victims [19], which has significant consequences on the future growth and development of a nation due to the reasons mentioned above. Furthermore, RTCs have been a prime cause of death among children and teenagers. Useche et al. [20] attributed this trend to the risky behavior of road users from the young age category.

Therefore, in this research, we focus on investigating factors related to the involvement of young people in RTCs. Thus, the purpose of this study is to investigate the interrelationships between various elements generated by inattentive driving behavior of the productive age group and the subsequent repercussions that result in RTCs. It would be easier to design appropriate road safety policies that would avoid RTCs if the impact and extent of incorrect driving attitudes and behaviors on the severity and types of traffic crashes could be detected. Saudi Arabia, similar to many other nations throughout the world, has developed methods and scenarios to aid in the mitigation and resolution of RTCs. However, despite the deterrent and awareness measures taken by the Traffic Department and other concerned departments to combat this danger, Saudi Arabia continues to have a significant problem with traffic crashes. The avoidance of traffic crashes, as well as their disastrous consequences, should begin with well-thought-out planning and scientific approaches. As a result, it is critical to examine the impact of driving behavior and to address the effects of driving attitude, driver education, and awareness—particularly of young drivers—on road traffic crashes, and to use cutting-edge technology solutions to improve road safety in the Saudi Arabian context. As evident from the above discussion, research related to the involvement of young people in RTCs is extremely important due to its importance and widespread occurrence.

The current study is devoted to analyzing the productive age group's (15 to 44 years) involvement in severe traffic crashes in Al-Ahsa, Saudi Arabia, by considering crash data for 2015–2018. It aims to investigate the factors significantly affecting the occurrence and severity of RTCs involving young road users in Saudi Arabia. Currently, there is a lack of literature related to this aspect of road traffic crashes that applies statistical and machine

learning methods, especially for Saudi Arabia and other countries in the region. Binary logistic regression and CART models were used for predicting the chances of involvement of young victims, ranging from 15 to 44 years of age, in RTCs in this study.

The remainder of this paper is organized as follows: In Section 2, we present an overview of the latest literature pertaining to this area of research and the importance of the study for Saudi Arabia; in Section 3, we elaborate on the source and nature of data and the modeling techniques; in Section 4, we provide the results and discussion of crash data; lastly, in Section 5, we provide a short account of findings from this study and recommendations for authorities and future research.

## 2. Literature Review

There is a body of research pertaining to RTCs that has been carried out to investigate and mitigate the factors related to RTCs. Experts in this field have focused on different aspects related to RTCs, including causal factors [21], severity analysis [22], and specific road users [23]. RTCs are caused by a variety of factors, including human driver error, vehicle characteristics, and traffic infrastructures, such as engineering design, road maintenance, and traffic regulation [24]. Driver attitude, including road courtesy and behavior; driving while under the influence of drugs, particularly alcohol; gender; seat belt use; and driver age (teenage drivers and elderly drivers) are among the recognized human factors associated with RTCs [24,25]. Touahmia (2018) [26] found that human factors are the prime cause of RTCs in Saudi Arabia, with 67% involvement. Among them, speed limits and seatbelt violations have been commonly reported among drivers involved in these traffic crashes.

There is a significant body of research that has been conducted to explore different aspects of driving behavior of young aged drivers around the world. Jonah [27] examined the evidence for the hypotheses that young (16–25) drivers were (a) more likely than older drivers to be involved in a fatal crash and (b) that this increased risk was mostly due to their proclivity to take risks when driving. Even after correcting for differences in the quantity and quality of road travel and driving experience, epidemiological studies have supported the first theory. Observational and self-report studies of driving behavior have supported the second hypothesis.

Doherty et al. [28] studied the situational dangers that young drivers face, particularly in terms of the passenger effect, using data sources provided by the Ontario Ministry of Transportation, Canada, provided in 1988. They estimated crash involvement rates by the number of passengers, time of day, and weekday. In all conditions studied, the crash participation rates of 16–19-year-old drivers were higher than those of 20–24- and 25–59-year-old drivers. However, they were significantly higher on weekends, at night, and with passengers. The effect of the passenger variable on the results is particularly intriguing since, unlike weekends and nights, passengers had a negative impact on total crash rates for 16–19-year-old drivers. This effect was noticeable in both sexes, with crash involvement rates around twice as high with passengers as without. Crash rates for 16–19-year-olds were also significantly higher for two or more passengers as compared with one passenger. Passengers in this age bracket experienced the greatest rates at night.

Chliaoutakis et al. [29] conducted research in Greece to determine and clarify the (possible) link between young drivers' lifestyles and the likelihood of being involved in a traffic crash. Factor analysis and logistic regression analysis were part of the statistical investigation. According to the logistic regression analysis, young drivers with a dominant lifestyle trait of alcohol use or driving without a destination had a higher crash risk, but those with a prominent lifestyle trait of culture had a lower crash risk. Furthermore, young drivers who were religious in some way appeared to be less likely to be involved in a crash.

To assess crash rates, Chen et al. [30] examined data from passenger vehicle crashes involving drivers aged 17–25, from 1997 to 2007, in New South Wales (NSW), Australia. Age, gender, rurality of location, and socioeconomic position were all taken into account when examining crash patterns over time by the severity of driver injury. From 1997 to

2007, the noninjury and fatality rates for young drivers declined by an average of 4% and 5% per year, respectively. From 2001 to 2004, young driver injury rates climbed by roughly 12% before declining. Males' relative collision risk (independent of driver injury) reduced dramatically over time as compared with that of females. Drivers aged 17 and, in particular, 18 to 20 years old had significantly and continuously greater crash risks than drivers aged 21–25 years old throughout the study period.

Al-Hemoud et al. [31] focused on motorists aged from 25 to 35 years old. The researchers wanted to see if there was a link between a man's style of living and his risk of being involved in a car crash. The findings showed that on Kuwait's countrywide public highways, motorists maintained insufficient space between their vehicles and those ahead of them, indicating dangerous driving behavior. Speeding has been found to be the most important predictor of traffic collisions.

Simons-Morton et al. [32] investigated psychological and personality variables of observed speeding among teenage drivers in the metropolitan area of Virginia in the USA. Speeding was linked to greater g-force event rates (r = 0.335, pb0.05), which increased with time and were predicted by day vs. night excursions, higher sensation seeking, drug use, deviance tolerance, peer pressure sensitivity, and the number of risk-taking friends. The relationship between speeding and risk-taking friends was significantly mediated by perceived danger.

Carpentier [33] studied to see how much of a role young novice drivers' familial environment had in their driving behavior. A group of young rookie drivers (N = 171) aged 17 to 24 years old who had held their permanent (or temporary) driver's license for less than a year took part in an online survey designed and circulated by the Hasselt University, Belgium. The survey asked about the participants' family climate, three socio-cognitive factors (attitude, locus of control, and social norm), and risky driving habits; both the family environment and socio-cognitive variables were predicted to have a direct impact on hazardous driving. The findings corroborated the significance of the three socio-cognitive variables, as attitude, locus of control, and social norm all strongly predicted self-reported hazardous driving.

Hassan [34] used structural equation modeling (SEM) to investigate the driving behavior of 18- to 24-year-old male Saudi motorists in Riyadh, the capital city of Saudi Arabia. The study looked at the factors that influence the involvement of young Saudi motorists in traffic crashes. The findings revealed that exceeding the stated speed limit was the most common reason for young Saudi motorists receiving traffic penalties (73%). In addition, "being late" was the leading cause of risky driving (62%).

Ramisetty-Mikler and Almakadma [35] conducted a survey on the attitudes of teenage motorists in Riyadh, as well as their risky driving behavior. Approximately 40% of respondents admitted to engaging in unsafe driving behavior known as "drifting". A total of 70% of drivers thought of "drifting" as a unique ability or a popular action. It was found that motorists were happy about engaging in risky behavior even though they knew it was dangerous.

According to a study conducted by (Issa) [36] in Tabuk city, Saudi Arabia, motorists under the age of 30 were involved in almost 60% of all RTCs, and more than 80% of all RTCs were caused by human factors. Drivers with advanced driving experience and scholastic achievements were involved in more traffic crashes than drivers without advanced driving and educational expertise.

Mohamed and Bromfield [37] used structural equation modeling (SEM) to investigate the links between road traffic crashes, driving behavior, and underdeveloped male motorists' views of road traffic safety in the Eastern Province of Saudi Arabia. A total of 287 drivers between the ages of 18 and 24 were included in the study. The findings revealed that the driving behavior of young Saudi male drivers could be divided into three categories: error-making, aggression, and neglect. Aggressive and negligent actions, unlike errors (violations), were both heavily influenced by drivers' attitudes toward road traffic safety, and both enhanced the probability of road traffic collisions.

Weston and Hellier [38] investigated the impact of peer influence on young drivers. They looked at the link between peer influence vulnerability and unsafe driving behavior among teenage drivers. According to their findings, young drivers who were persuaded by their peers to gain social status and who had peers intervene in their decisions committed more driving offenses.

Zeyin et al. [39] looked at the effect of a safe driving environment among friends on prosocial and aggressive driving behaviors among young Chinese drivers and concluded that traffic locus of control had a moderating function. A total of 352 young Chinese drivers, ranging in age from 18 to 25, volunteered to take part in the study and filled out a questionnaire that included questions about safe driving atmosphere among peers, traffic locus of control, and prosocial and aggressive driving behaviors. Prosocial and aggressive driving habits were directly influenced by the driving atmosphere among friends and the traffic locus of control. A comparative summary of papers on young drivers' behaviors is shown below in Table 1.

**Table 1.** Comparative summary of papers on young drivers' behaviors.

| References | Focus Group | City/Country | Comments |
|---|---|---|---|
| Brian A. Jonah [27] | 16–25 years | Canada | Successfully tested two hypotheses: (a) young drivers are more likely to be involved in a fatal crash than older drivers and (b) that this higher risk is mostly due to their tendency to take risks when driving. |
| Sean T. Doherty et al. [28] | 16–19 years, 20–24 years, 25–59 years | Ontario, Canada | Studied the situational dangers that young drivers of different age groups face. Crash rate for the 16–19-year-old age group was higher than other age groups. The rate of traffic crashes was even higher on weekends and nights. |
| Joannes El. Chliaoutakis et al. [29] | 18–24 years | Greece | Investigated the potential relationship between the lifestyle of teenage drivers and their chances of becoming involved in a traffic crash using factor analysis and logistic regression analysis. |
| H. Y. Chen et al. [30] | 17–25 years | NSW, Australia | Examined passenger vehicle crash pattern over time by severity of driver injury considering age, gender, rurality of location, and socioeconomic position. Drives aged 17–20 years had significantly and continuously greater risk than drivers aged 21–25 years old. |
| Al-Hemoud et al. [31] | 25–35 years | Kuwait | Examined the relation between living style and crash risk-taking behavior of drivers. |
| Bruce G.Simons-Morton et al. [32] | Teen ages | Virginia Metropolitan Area, USA | Investigated psychological and personality variables of observed speeding. The relationship between speeding and risk-taking friends was significantly mediated by perceived danger. |
| Aline Carpentier [33] | 17–24 years | Hasselt, Belgium | Studied the influence of family environment and socio-cognitive elements on driving behavior on young novice drivers; both family environment and socio-cognitive variables were predicted to have a direct impact on hazardous driving. |



**Table 1.** *Cont.*

| References | Focus Group | City/Country | Comments |
|---|---|---|---|
| Hassan [34] | 18–24 years | Riyadh, Saudi Arabia | Studied the factors that affect the involvement of young Saudi motorists in traffic crashes, where findings revealed that more than 70% of penalties and 60% of risky driving were due to exceeding the posted speed limit and "being late", respectively. |
| Ramisetty-Mikler and Almakadma [35] | Teen age | Riyadh, Saudi Arabia | Conducted a survey on teenage drivers, where it was found that 40% of drivers "drift" cars as an act of adventure despite knowing that it is a hazardous action. |
| Issa [36] | Below 30 years | Tabuk, Saudi Arabia | Showed that experienced and educated drivers were more involved in RTCs than less experienced and educated drivers. |
| Mohamed and Bromfield [37] | 18–24 years | Eastern Province, Saudi Arabia | Young Saudi male drivers were divided into three categories: (i) error-making, (ii) aggressive, and (iii) negligent. |
| Lauren Weston and Elizabeth Hellier [38] | Teen age | USA | Studied the relationship between peer influence vulnerability and unsafe driving behavior and suggested peer education tools for safe driving. |
| Yang Zeyin et al. [39] | 18–25 years | China | Investigated the influence of a safe driving environment among friends on prosocial and aggressive driving behavior. |

In recent years, there has been an increase in research on RTCs involving vulnerable road users (VRUs). RTCs have been the leading cause of death in the productive (working) age group (15–44 years), and the present study focused on this age group [7]. This category includes all road users including VRUs, such as drivers, pedestrians, passengers, and cyclists. In developing countries, road traffic injuries primarily affect pedestrians, passengers, and cyclists, as opposed to drivers, who account for the most deaths and disabilities in the developed world. In the United States, for instance, drivers account for more than 60% of road traffic crash fatalities, whilst drivers account for less than 10% of mortalities caused by traffic injury issues in the least-motorized countries [8,9].

However, most previous research on pedestrian safety and traffic crashes have been crash-based studies, and researchers have looked into the characteristics of older pedestrians, the living and traveling environment, and roadway features associated with pedestrian crashes. The authors in [40], for example, used various data-mining algorithms, including the classification and regression tree (CART) model, gradient boosting (GB) model, random forest (RF) model, artificial neural network (ANN) model, and support vector machine (SVM) model, to ascertain the most important factors that contribute to death and severe vehicle–pedestrian crashes at intersections. In another study, the component-wise, model-based, gradient-boosting algorithm was used to estimate the nature and effects of socioeconomic, land use, road network, and traffic features on pedestrian crashes in Broward and Miami-Dade Counties in Florida [41]. The XGBoost machine learning tool was used to simulate the problem of classifying three categories in older pedestrian traffic crashes [42]. Another study provided a framework for using machine learning Bayesian neural network (BNN) approaches to reduce pedestrian deaths due to traffic crashes [43]. Pour et al. [44] used decision tree (DT) and kernel density estimation (KDE) to investigate the effects of temporal, geographical, and personal variables on the severity of vehicle–pedestrian collisions. Ding et al. [45] used multiple additive poisson regression trees (MAPRT) to provide a different perspective on the effects of pedestrian collisions and SVM, as well as multinomial logit (MNL) were used to forecast the severity of injuries in pedestrian collisions [32,46]. Zou [47] proposed a model for predicting vehicle acceleration based

on machine learning and driving behavior analysis. Two different groups were considered. The first group consisted of 10 homogeneous drivers, whereas the second group consisted of 20 heterogeneous drivers. The driving behavior semantics were divided using a finite mixture of the hidden Markov model (MHMM). The Kolmogorov–Smirnov test was used to assess the similarity of distinct behavioral semantic chunks, and both groups employed long short-term memory (LSTM) and the gate recurrent unit (GRU) to forecast vehicle acceleration.

It can be seen that various machine learning algorithms and models have been applied for various aspects of crash severity analysis. There is, however, a significant research gap. First, no studies with comparative analysis of predictive age group crash severity using machine learning modeling were found. Second, in existing machine-learning-based crash severity modeling, a confined number of road traffic crash variables were considered as input features. As a result, it would be worthwhile to conduct a thorough analysis and compare various road traffic crash parameters based on machine learning modeling for the productive age group.

The present study is based upon RTC data from Saudi Arabia, which is a car-dependent country, which has caused it to suffer problems associated with RTCs. Despite the harsh penalties enforced by the General Traffic Department for lawbreakers, road traffic crashes cause a considerable number of deaths each year in Saudi Arabia. A study reported an 8% increase in RTC-related mortalities in Saudi Arabia between 2005 and 2010 [48]. Traffic crashes are one of the most serious challenges that society and its stakeholders face due to various variables and contributing factors. Because it is a sophisticated behavioral problem of young drivers, it involves many stakeholders. To the authors' best knowledge, a study of this nature is not found in the literature for Saudi Arabia and other countries in the region. It is expected that the outcomes of this research will help researchers to improve their understanding of the involvement of young people in RTCs. It will also help traffic management authorities, especially in Saudi Arabia, to minimize RTCs, especially those involving young and productive people, thereby mitigating their impacts on the country's future. In 2018, Saudi Arabia allowed female citizens to have a license, which increased the number of young drivers in the country. Car ownership and RTCs are also expected to increase [49]. Hence, it is crucial to proactively mitigate RTCs involving young road users in Saudi Arabia.

## 3. Study Area and Data Sources

The case study area for this study is Al-Ahsa city in Saudi Arabia's Eastern Province (Figure 1). Because of the high crash rate mentioned in previous research, Al-Ahsa was chosen for this investigation. Between 2009 and 2016, in the Eastern Province, 31.9 percent of all traffic crashes occurred in Al-Ahsa, followed by Dammam, Hafr Al-Batin, Jubail, Qatif, Dhahran, and Khobar, sequentially. Other cities had crash rates of less than 5% [50]. This clearly reveals that among the cities in the Eastern Province, Al-Ahsa is the most prone to traffic crashes. Al-Ahsa has recently been recognized as one of Saudi Arabia's UNESCO-listed heritage sites and has also been given a Guinness World Record, which means that the city has tremendous potential as an international tourist attraction [51]. However, the city's high crash rate may jeopardize that possibility, necessitating immediate action. The Traffic Police Department in Dammam provided the crash data for Al-Ahsa used in this study from October 2014 to May 2018. Overall, the dataset shows that collisions between cars were the most common type of crash, accounting for about 8% and 30% of fatal and injury events, respectively. Vehicle overturning was another common form of crash, accounting for 6.5 percent of fatal and injury incidents, respectively. Furthermore, pedestrians' vulnerability was demonstrated by the fact that 12.5 percent of injury traffic crashes and 2% of deadly traffic crashes involved pedestrians. Some crash types, including collisions with road guardrails, motorcyclists, parked cars, and permanent objects, provide a risk of harm, despite fatal crash rates being low (Figure 2). Further analysis and a

description of the dataset are provided in Section 4. A sample of the dataset can be seen in Appendix A.

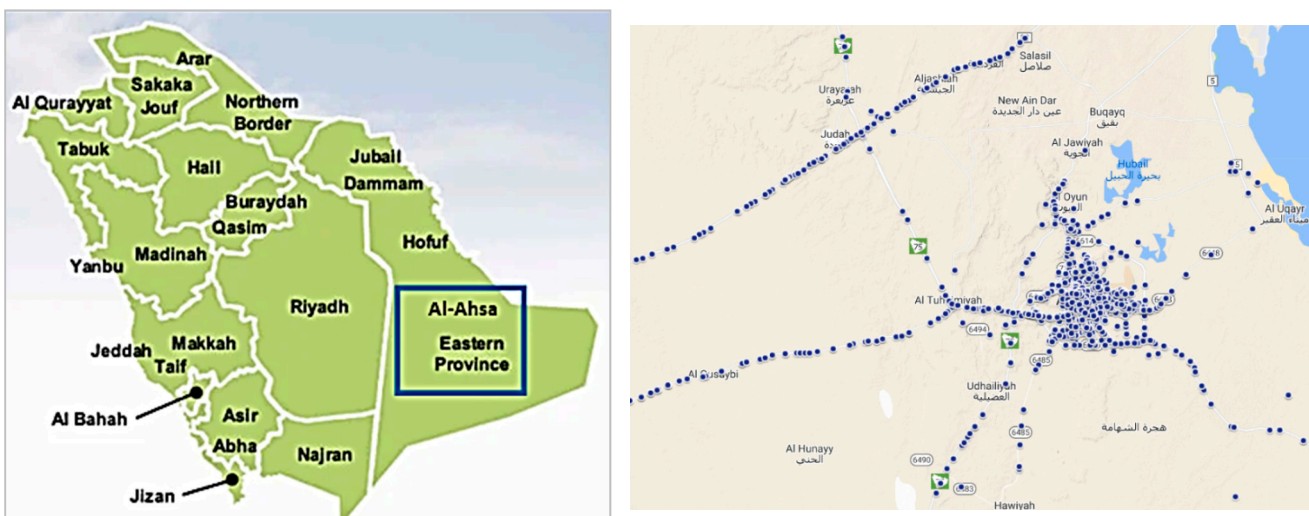

**Figure 1.** Study area and locations of traffic crashes in the Al-Ahsa region of Saudi Arabia (adopted from [51]).

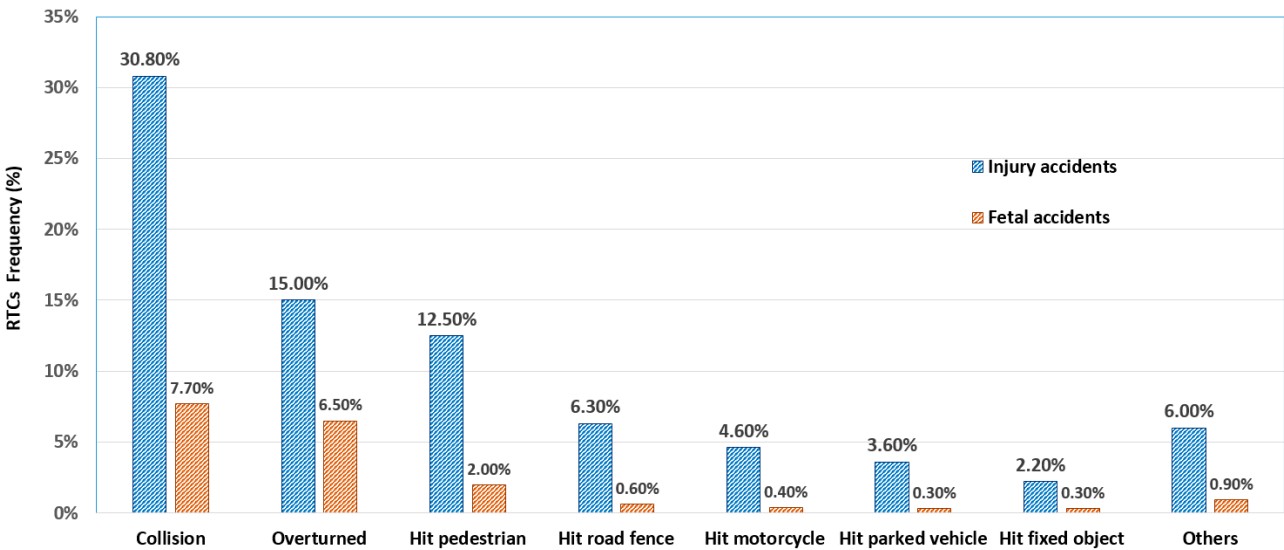

**Figure 2.** Frequency distribution of RTCs with types of crashes in the study area (adopted from [51]).

*Modeling Techniques*

Two types of models were used in this research to predict the involvement of people of the target age group, namely, binary logistic regression and classification and regression tree (CART) models. These models are widely used for classification problems in different fields of transportation research, including road traffic crashes. The models were developed to predict the binary response variable if a victim from a certain age category was involved in the traffic crashes or not. More details about the selection of age categories are presented in the next section.

The logistic regression model works on the principle of maximum likelihood, as shown in Equation (1). A utility function is developed with a linear functional form, as per Equation (2). The utility function is subjected to a logistic function to find the probability of the particular outcome (see Equation (3)).

$$Model\ likelihood = Max[\sqcap P_i(n)] \tag{1}$$

$$U_n = b + \sum_{j=1}^{I} Y_j X_j \tag{2}$$

$$P_i(n) = 1/(1 + e^{-U_n}) \tag{3}$$

where $P_i$ is the probability of ith sample in the dataset (of "$I$" values) to have the outcome "$n$", which is the actual outcome; $U_n$ is the utility function for outcome $n$ (from "$N$" possible outcomes); $b$ is the intercept for the function; $X$ is an array of "$J$" independent variables; and $Y$ is an array of coefficients pertaining to each variable [52]. Logistic regression models have common usage in the prediction of categorical responses, and they have been used in predicting crash severity [53].

CART is a type of artificial intelligence technique that works on the principle of finding a suitable breakdown of input space to maximize the probability of the correct outcome. This is achieved through minimization of the Gini index, which is shown in Equation (4), at each node to divide it into child nodes. The parent node indicates the most influential variable at a particular level in the tree, while its child node represents the best division or split of that variable [54].

$$G = \sum (p_k(1 - p_k)) \tag{4}$$

where $p_k$ is the probability for the outcome $k$. The CART model is a non-parametric technique that is commonly applied for classification problems. It has also been used in predicting crash severity [55].

## 4. Analysis and Results

Firstly, a description of the general trends found in the RTC dataset is presented in the proceeding section. Then, an analysis related to young victims of traffic crashes is discussed.

### 4.1. Trends in RTCs

The dataset used for the analysis comprised 4093 traffic crashes that occurred from 2015 to 2018. There were 9031 victims in the traffic crashes. A total of 19% of the RTCs included fatalities, while the remaining had serious injuries. The primary reasons for traffic crashes included sudden turning, speeding, and not giving way, which accounted for approximately 47%, 18%, and 15% of traffic crashes, respectively. The chronological trend of traffic crashes shows a decrease in fatal and serious injury traffic crashes within the available dataset, as shown in Figure 3. The authors of [56], which was also related to Saudi Arabia, attributed the decrease in severe traffic crashes to the installation of traffic cameras in their study.

A seasonal and month-wise analysis of fatal and severe injury traffic crashes is shown in Figures 4 and 5, respectively, which also confirm the trend that the number of RTCs shows a general decreasing trend, between 1436 and 1438, on the Hijri calendar.

A one-way ANOVA was performed to check the following null hypothesis: Lunar months do not have a significant effect on the occurrence of RTCs.

The results of the ANOVA are shown in Table 2. It is shown in the table that the Fisher's statistic (F-value) for the ANOVA has a probability ($p$-value) of 0.5, indicating a very low value of the statistic. The F-value is considered to be statistically significant if it has a probability of 0.05 or less, as per the standard criterion for statistical tests [57]. Hence, in this case, the abovementioned null hypothesis can be accepted.

**Table 2.** ANOVA tests for lunar months.

| | Sum of Squares | Degrees of Freedom | Mean Sum of Squares | F-Value | *p*-Value |
|---|---|---|---|---|---|
| **Intercept** | 444,889 | 1 | 444,889 | 407 | 0.00 |
| **Month** | 11,496 | 11 | 1045 | 0.95 | 0.51 |
| **Error** | 26,239 | 24 | 1093 | | |

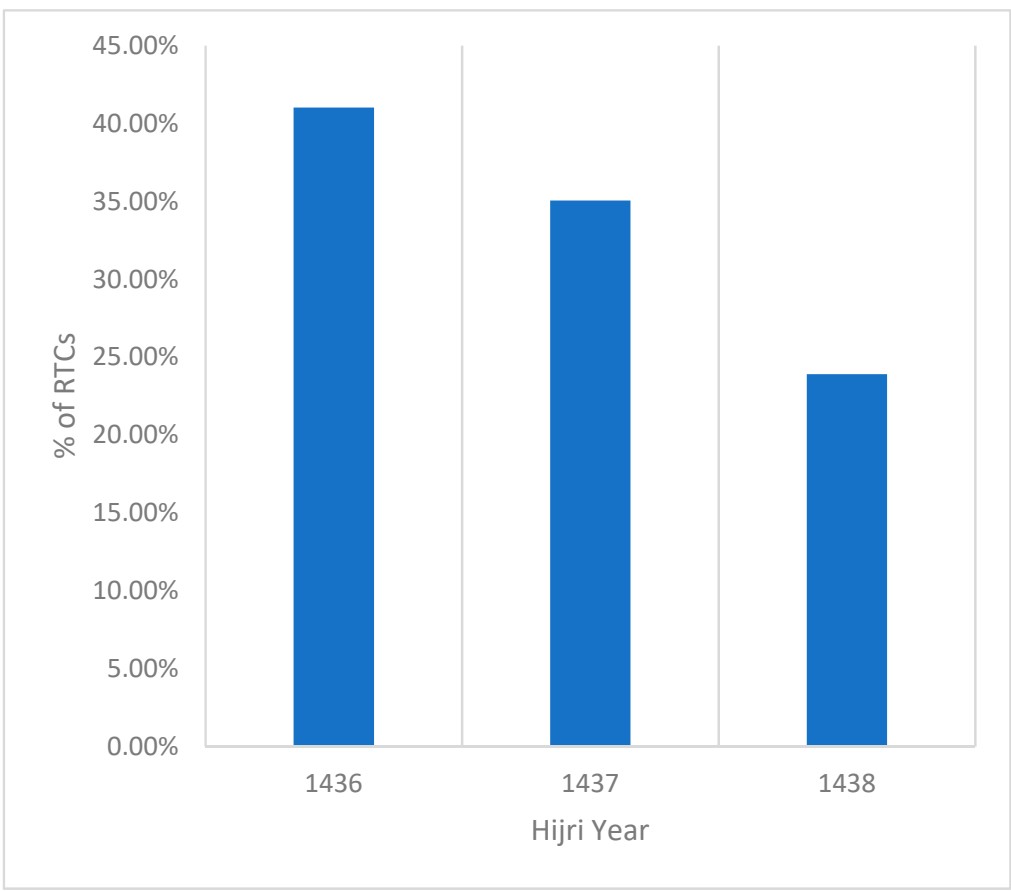

**Figure 3.** Chronological trend of fatal and serious injury traffic crashes.

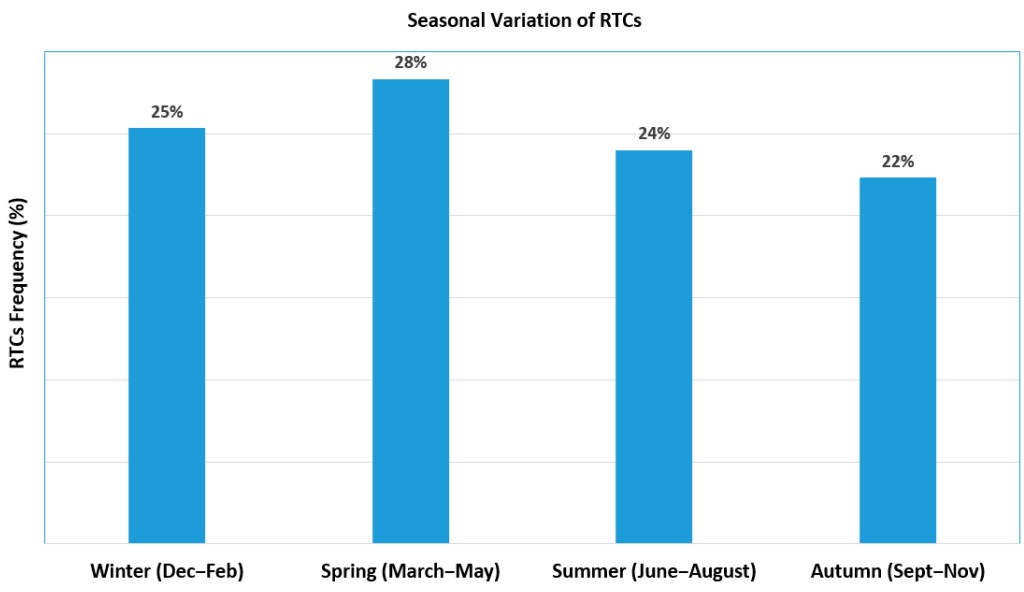

**Figure 4.** Seasonal variation of RTCs in the study area [51].

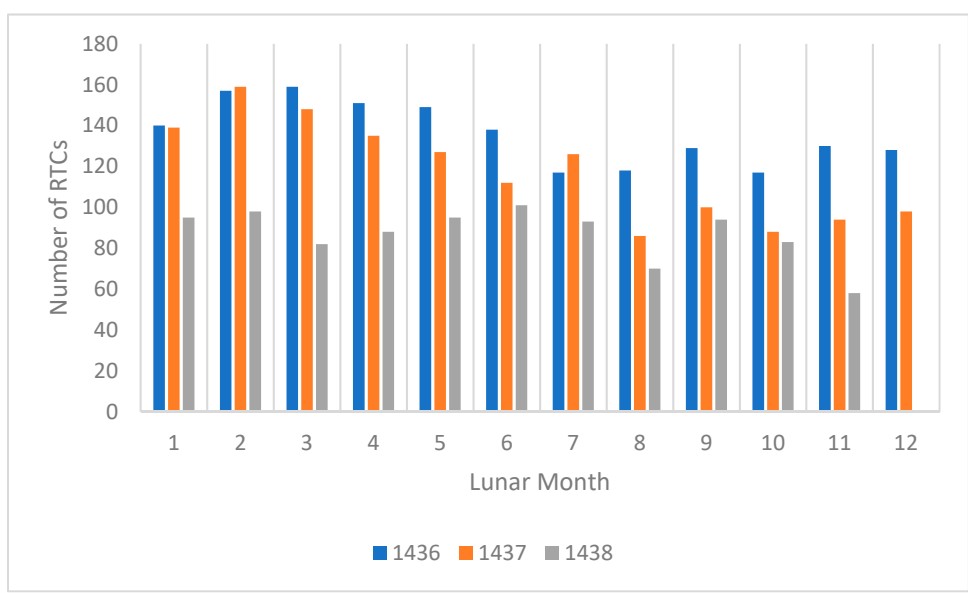

**Figure 5.** Month-wise distribution of fatal and severe injury traffic crashes.

The crash type "fatal and severe injury RTCs involving a moving vehicle" was higher than any other type of crash in the available dataset and was approximately 38%. Other prominent types of traffic crashes included those involving multiple vehicles (21% of RTCs) and running off the road (14% RTCs). The authors of [50] also reported the first two as common types of traffic crashes reported in the Eastern Province. The study area from which data were collected for this research is a part of this province of Saudi Arabia.

Table 3 provides some descriptive statistics related to the number of victims of major injuries and deaths in RTCs within the available dataset. The severity index is very high, showing that almost all RTCs have at least one death victim, on average. The population of the Al-Ahsa area is reported to be 1.3 million [58]. The figures shown in Table 3 are high as compared with other studies found in the literature. The authors of [59] showed an annual rate of merely 3.3 for fatal traffic crashes in Iowa, USA, as compared with 352 in Al-Ahsa for the available dataset. In another study, ref. [60] reported 14.93 deaths per 100,000 people in Columbia, while it was 311 for the study area. Interestingly, when the annual rates for the occurrence of crash rates were compared with previous studies, they were not found to be higher for the study area. In a study completed by the authors of [59], the annual fatal crash rate was approximately 350, which is close to the value for the study area. The authors of [61] did not mention the fatal and major injury RTC rates per year, but this point was further validated by the following studies. On the one hand, Abdi et al. [62] reported a rate of 1463 per year for fatal and major injury RTCs for Addis Ababa, while [61] reported it to be 3631 per year for Torino (Italy). On the other hand, the study area had a rate of 1333 fatal and major injury RTCs per year. This points to the fact that RTCs in the study area, even with lower occurrence rates, are more severe and involve more victims than in other parts of the world.

An age-wise distribution of victims is presented in Figure 6. The distribution shows that the most vulnerable age group is between 15 and 44 years, which has a higher frequency of traffic crashes than other age groups. The data show that more than 73% of the traffic crashes involve victims in this age group, i.e., 6641 out of 9031 traffic crashes. A similar age group was reported to be commonly involved in RTCs for developed countries by [63]. People within this age group are considered to be the most productive part of society [31,64]. Due to these reasons, further analysis in the form of prediction models was conducted and focused on victims of this age group.

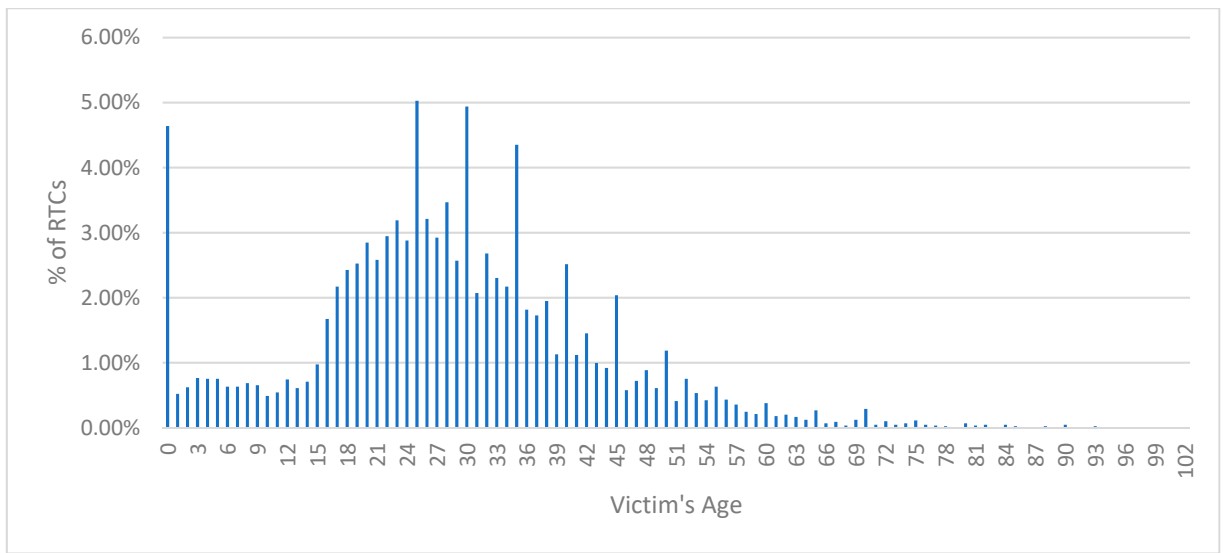

**Figure 6.** Age-wise distribution of RTC victims.

**Table 3.** Descriptive statistics for RTCs, major injuries, and deaths.

| Parameter | Number of Major Injuries | Number of Deaths |
|---|---|---|
| Total | 22,787 | 3579 |
| Number of RTCs | 3245 | 756 |
| Average per RTC * | 7 | 4 |
| Range | 0–46 | 0–13 |
| Severity Index = deaths/no. of RTCs | 0.89 | |
| Number of fatal and major injury RTCs per year | 1333 | |
| Number of fatal RTCs per year | 352 | |
| Number of deaths per year | 1193 | |
| Number of deaths per 100,000 people | 311 | |

* Number of major injuries/number of RTCs with major injuries; same was used for the average deaths per RTC.

### 4.2. Prediction Models

Binary logistic regression and CART models were used for predicting the chances of young victims, ranging from 15 to 44 years of age, being involved in RTCs. For the development of the models, crash types and reasons with less than 5% occurrence in the dataset were merged into a single category of "others". As stated above, 5% is the commonly applied significance criteria for statistical analysis. The parameters taken for modeling and their description/coding are provided in Table 4. The coding for crash reasons and types was kept the same as that of the original dataset for backtracking purposes. The composition of model development–validation samples was 75–25%. It was found that approximately 74% of the samples included victims between the ages of 15 and 44 years. Hence, we ensured that samples that were selected for development and validation were of the same proportion for the accurate representation of actual data. The number of samples was selected randomly within each category of RTCs.

**Table 4.** Description of modeling parameters.

| Parameter | Classes | Description/Code |
|---|---|---|
| Severity | Fatal | 1 |
| | Major injury | 2 |
| Crash type | Collision | 1 |
| | Hit motorcycle | 3 |
| | Hit road fence | 6 |
| | Hit pedestrians | 8 |
| | Vehicle overturn | 9 |
| | Other types | 20 |
| Crash reason | Driver distraction | 3 |
| | Speeding | 4 |
| | Not giving way | 9 |
| | Sudden turning | 14 |
| | Not leaving sufficient distance | 15 |
| | Other reasons | 46 |
| Number of deaths | N/A | Number of persons killed in RTC |
| Number of injuries | N/A | Number of persons injured in RTC |
| Age | 15 to 44 (inclusive) | 1 |
| | Less than 15 or more than 44 | 0 |

Equation (5) shows the utility equation for the logistic regression model. The number of deaths was not found to have a significant effect on the model at a 5% significance level. Equation (5) is:

$$U = -1.03 + 0.08 \text{ (fatal)} - 1.64 \text{ (collision)} + 0.01 \text{ (vehicle overturn)} + 0.60 \text{ (hit pedestrians)}$$
$$-0.08 \text{ (other types)} - 0.002 \text{ (hit road fence)} - 0.30 \text{ (not leaving sufficient gap)}$$
$$-0.18 \text{ (sudden turning)} + 0.04 \text{ (speeding)} + 0.25 \text{ (not giving way)} + 0.14 \text{ (other reasons)}$$
$$+0.04 \text{ (number of injuries)}$$

$$(5)$$

The *t*-statistics and corresponding *p*-values for the coefficients of the utility equation are given in Table 5. It can be observed that some of the reasons and types of traffic crashes did not have a statistically significant coefficient in the model, for example, vehicle overturn, hit road fence, and any other types of traffic crashes; however, they were kept in the equation because they were fed to the model under the categorical variables of reasons and types. This strategy was chosen to reduce the number of variables used for the model. Consequently, it was not possible to exclude specific categories of the same variable. However, after the first trial, as shown in Equation (5), we decided to include a separate binary variable for each reason and type of crash and recalibrate the utility equation by omitting the reasons and types that did not have a significant effect on the previous utility equation. The utility equation, acquired in this manner, is shown as Equation (6) as follows:

$$U = 1.07 + 0.84 \text{ (hit pedestrians)} - 0.40 \text{ (not leaving sufficient gap)} - 0.23 \text{ (sudden turning)}$$
$$+0.04 \text{ (number of injuries)} + 0.09 \text{ (fatal)}$$

$$(6)$$

**Table 5.** The *t*-statistics and *p*-values for the coefficients of the utility Equation (1).

| Variable | *t*-Statistic | *p*-Value |
|---|---|---|
| Intercept | −19.26 | 0.00 |
| Fatal | 2.25 | 0.02 |
| Collision | −2.78 | 0.01 |
| Vehicle overturn | 0.15 | 0.88 |
| Hit pedestrians | 7.46 | 0.00 |
| Other types | −1.08 | 0.28 |
| Hit road fence | −0.02 | 0.98 |
| Not leaving sufficient gap | −3.16 | 0.00 |
| Sudden turning | −3.35 | 0.00 |
| Speeding | 0.57 | 0.57 |
| Not giving way | 3.12 | 0.00 |
| Other reasons | 1.64 | 0.10 |
| Number of injuries | 5.45 | 0.00 |

The *t*-statistics and *p*-values for the utility Equation (6) are given in Table 6. It can be observed that traffic crashes that result in hitting pedestrians have a higher likelihood of involving people from the target age group. Since no other crash type significantly affected the utility equation, it can be said that all other crash types have similar chances of involving people from or outside of the target age group. Moreover, traffic crashes due to not leaving a sufficient gap and sudden turning are less likely to involve victims from the target age group, while all other reasons had similar chances of their involvement. Another important observation is that the traffic crashes involving the target age group are more severe and have a higher number of victims, which is shown by the positive coefficients of the number of injuries and fatal traffic crashes.

**Table 6.** The *t*-statistics and *p*-values for the coefficients of the utility Equation (6).

| Variable | *t*-Statistic | *p*-Value |
|---|---|---|
| Intercept | −19.26 | 0.00 |
| Hit pedestrians | 2.25 | 0.02 |
| Not leaving sufficient gap | −2.78 | 0.01 |
| Number of injuries | 0.15 | 0.88 |
| Fatal | 7.46 | 0.00 |

Values for accuracy, precision, error, recall, false discovery rate (FDR), false-negative rate (FNR), and F1 score were also calculated for the logistic regression model, as per Equations (7)–(13) [65]. These metrics are shown in Table 7 for development and validation samples.

$$\text{Accuracy} = (\text{TP} + \text{TN})/(\text{TP} + \text{FN} + \text{FP} + \text{TN}) \tag{7}$$

$$\text{Precision} = \text{TP}/(\text{TP} + \text{FP}) \tag{8}$$

$$\text{Recall} = \text{TP}/(\text{TP} + \text{FN}) \tag{9}$$

$$\text{Error} = 1 - \text{Accuracy} \tag{10}$$

$$\text{FDR} = 1 - \text{Precision} \tag{11}$$

$$\text{FNR} = 1 - \text{Recall} \tag{12}$$

$$\text{F1 Score} = (2 \times \text{Precision} \times \text{Recall})/(\text{Precision} + \text{Recall}) \tag{13}$$

**Table 7.** Accuracy measures for the logistic regression model.

| Parameter | Dataset | Value |
|---|---|---|
| Accuracy | Development | 0.73 |
| | Validation | 0.73 |
| Precision | Development | 0.74 |
| | Validation | 0.74 |
| Recall | Development | 1.00 |
| | Validation | 0.98 |
| Error | Development | 0.27 |
| | Validation | 0.27 |
| FDR | Development | 0.26 |
| | Validation | 0.26 |
| FNR | Development | 0.00 |
| | Validation | 0.02 |
| F-1 | Development | 0.85 |
| | Validation | 0.84 |

Here, TP refers to a true positive, which is a case in which the model predicted the victim to be in the target age group while he was actually in that category; TN refers to a true negative, which is a case in which the model predicted the victim to be out of the target age group while he was actually not in the target age group; FN refers to a false negative, which is a case in which the model predicted the victim to be out of the target age group while he was actually in that group; lastly, FP refers to a false positive, which is a case in which the model predicted the victim to be in the target age group when he was actually not in that group.

All accuracy measures for the logistic regression model are satisfactory, and they are close to each other for development and validation datasets. This shows that the logistic model is accurate and robust in predicting the age group of the crash victims.

The CART model developed for the crash dataset is shown in Figure 7, and its accuracy measures are shown in Table 8. The distribution of development and validation datasets was the same as the logistic regression model for a fair comparison. In Figure 7, each node shows the factor used to divide the node further. In the case of a terminal node (last node in that branch), the limiting value of the variable and the probability of the victim being in the target age group (*p*-value) are shown. The sequence of each node shows the conditions that would result in a certain probability of the involvement of the target age group. For example, hitting pedestrians results in a 57% probability that a victim from the target age group would be involved in the crash, irrespective of any other factor (node ID 2). Similarly, if the crash is of any other type and has more than six injuries, then there is a 60% probability of involving the target age group (node ID 4).

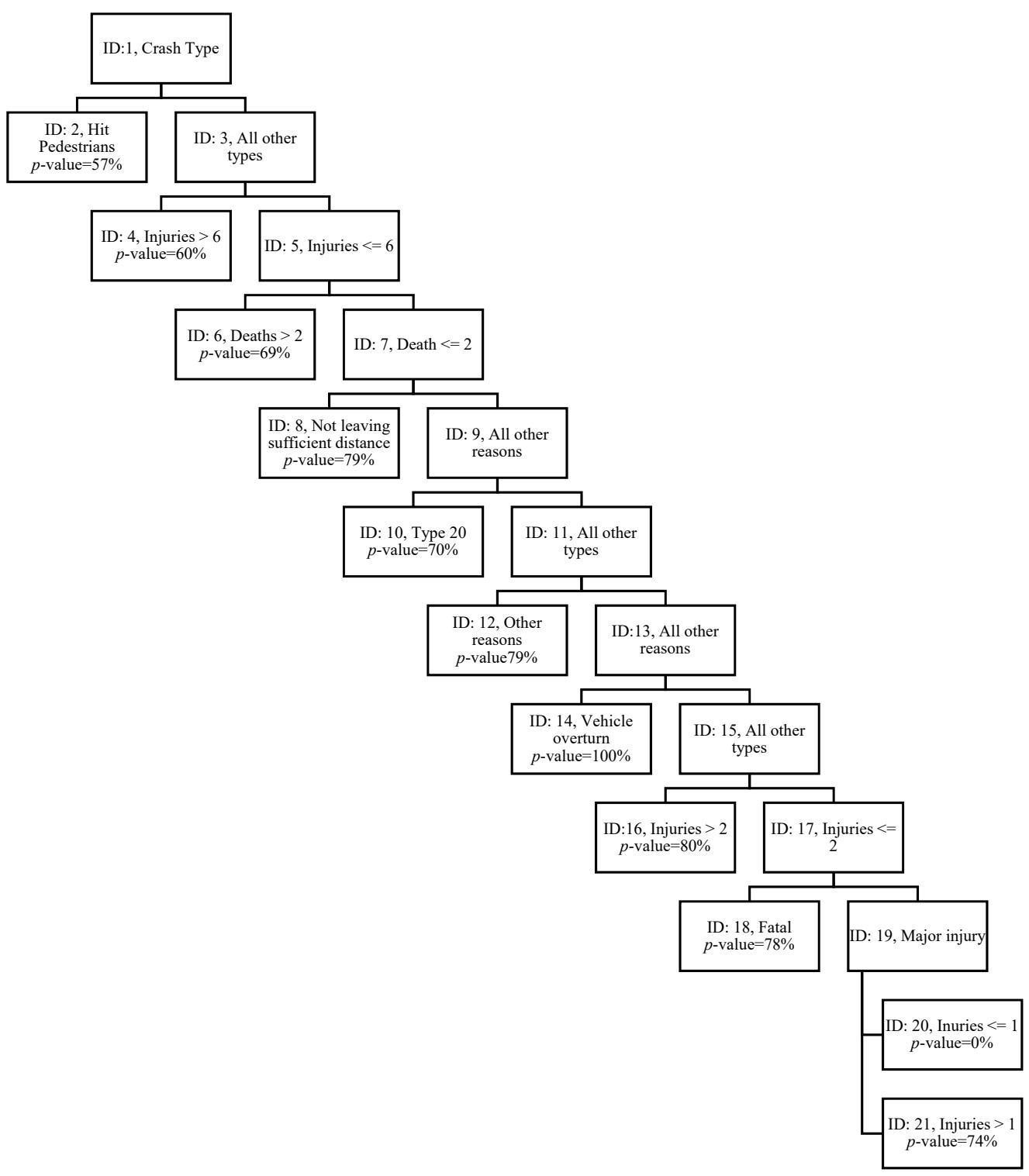

**Figure 7.** CART model.

As shown Figure 7, the highest probability for a victim in the target age group is when the crash was a vehicle overturn, it has less than or equal to two deaths and six injuries, and the crash reason is either driver distraction, speeding, not giving way, or sudden turning (ID 14). Similarly, the lowest probability for a victim in the target age group is when the crash was for the abovementioned reasons; it is non-fatal with only one injury, and the crash type is either collision, hit motorcycle, or hit road fence. Apart from these extreme cases, the CART tree also shows that pedestrian traffic crashes have a relatively

low probability of involving people from the target age group as compared with other traffic crashes (ID 2). This could indicate that traffic crashes for the target age group are not due to lost control over the vehicle. Similarly, fatal traffic crashes have a relatively higher probability of a victim from the target age group as compared with traffic crashes with only major injuries. Traffic crash types that are combined under "other types", comprising less than 5% of crashes, also have a comparatively lower probability of involving victims from the target age group. Traffic crashes that happen due to not leaving a sufficient gap have a 79% possibility of involving the target age group in all types, except hitting pedestrians, as shown in node ID 12.

**Table 8.** Accuracy measures for the CART model.

| Parameter | Dataset | Value |
|---|---|---|
| Accuracy | Development | 0.74 |
| | Validation | 0.74 |
| Precision | Development | 0.74 |
| | Validation | 0.73 |
| Recall | Development | 1 |
| | Validation | 1 |
| Error | Development | 0.26 |
| | Validation | 0.26 |
| FDR | Development | 0.26 |
| | Validation | 0.27 |
| FNR | Development | 0 |
| | Validation | 0 |
| F-1 | Development | 0.85 |
| | Validation | 0.84 |

Findings from other terminal nodes are as follows: Traffic crashes with fewer than six injuries and two deaths, which do not happen due to insufficient gaps and result in a vehicle overturn, have a 100% probability of involving victims from the target age group. Traffic crashes that do not happen due to insufficient gaps and do not result in a vehicle overturn or hitting pedestrians have an 80% chance of involving a victim from the target age group if the number of injuries in the crash is more than two, which is shown in node ID 16. Similarly, if the number of injuries is less than two, but it has at least one death, then the crash has a 78% probability of involving a victim from the target age group (ID 18). Node ID 21 shows that this probability reduces to 74% if the crash had two injuries, with at least one of them being a major injury.

Model parameters shown in Table 8 for the CART model can be considered to be satisfactory in light of previous research [66,67]. Moreover, they do not show any significant change between development and validation datasets. Hence, it could be said that the CART model can accurately predict the involvement of the target age group in traffic crashes. Moreover, it is robust and can give similar accuracy on new datasets.

*4.3. Comparison*

Accuracy measures of logistic regression and CART models for the validation datasets are shown in Figure 8. It can be observed that they are almost the same. Hence, one model does not have an advantage over the other in terms of accuracy.

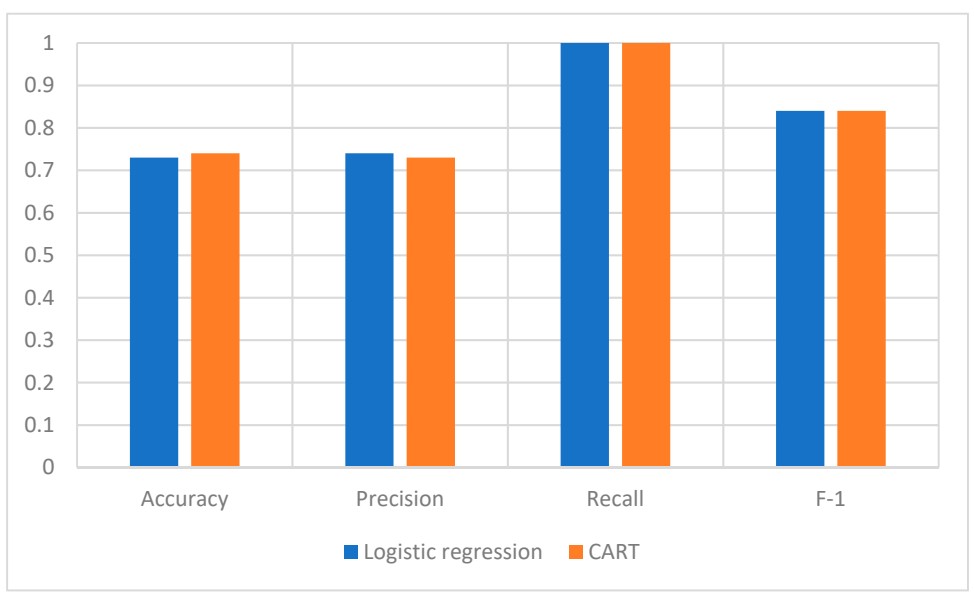

**Figure 8.** Comparison of prediction performance of logistic regression and CART models.

However, the CART model has several other advantages over the logistic regression model. First of all, variables for the reason and type of traffic crashes are not required to be converted into multiple binary variables because the CART model detects the best breakup of values/categories for each node. Secondly, the CART model does not have the statistical restriction of the significance of coefficients, therefore, all variables that have any impact on the outcome can be kept in the model. Lastly, the CART model is more efficient in displaying the interrelationships of independent variables. It can be seen from Figure 5 that variables for the reason and type of traffic crashes appear more than once in the tree, and their child nodes are linked with other variables, for example, the number of deaths. Therefore, one can see clearly which crash impact types/reasons are linked, and which ones are not linked, to the involvement of the target age group. This is not possible with the utility function provided by the logistic regression model, which is unable to consider interrelations among the independent variables. Another important factor for researchers is the time required to develop and run the model. In this regard, the CART model needs more time as compared with the logistic regression model. For the available dataset of this study, the processing times are 1 and 5 s for the logistic regression and CART models, respectively. It is expected that the difference in processing may be even higher for a larger dataset in favor of the logistic regression model.

## 5. Conclusions

This study aimed to investigate the involvement of people aged 15 to 44 in fatal and major RTCs in Al-Ahsa, Saudi Arabia. First, some descriptive analyses were completed to justify the need to analyze the target age group. Then, predictive models were developed using logistic regression and CART models to investigate the factors affecting the involvement of the target age group in RTCs.

The most common types of traffic crashes are collisions between two vehicles, followed by those involving multiple vehicles and pedestrians. Al-Ahsa city has a high severity index for RTCs in spite of having a lower frequency as compared with other cities. Logistic regression and CART models confirm victims from the target age group are involved in severe traffic crashes with comparatively higher numbers of injuries and fatalities. The CART model also shows overturn RTCs which happen due to driver distraction, speeding, not giving way, or sudden turning are more likely to involve victims from the target age group. The same reasons are also observed to give more than 75% probability of involvement of the target age group for other types of traffic crashes as well, except running

off the road. From these observations, it can be stated that traffic crashes involving the target age group mainly occur due to unsafe driving behaviors and can be reduced with the application of cautious driving practices. On the basis of findings of this research, it is recommended for future studies to investigate the impacts of changes in driving learning and teaching methodologies to eradicate irresponsible driving behavior, especially those which have been identified above. Other artificial intelligence techniques, such as deep learning, can also be utilized in such problems, which may provide better accuracy. The use of such techniques should be explored in future studies.

**Author Contributions:** Conceptualization, U.G., M.K.I. and M.A.; methodology and software, U.G. and M.K.I.; validation and formal analysis, U.G. and M.K.I.; resources and data curation, M.K.I.; writing—original draft preparation, review, and editing, M.K.I., U.G., M.A. and R.A.; project administration, M.K.I.; funding acquisition, M.K.I. All authors have read and agreed to the published version of the manuscript.

**Funding:** This work was financially supported by the Deanship of Scientific Research in the King Faisal University, Saudi Arabia (grant 267).

**Institutional Review Board Statement:** Not applicable.

**Informed Consent Statement:** Not applicable.

**Data Availability Statement:** The data that support the findings of this study are available from the corresponding authors Md. Kamrul Islam (maislam@kfu.edu.sa) and Uneb Gazder (ugazder@uob.edu.bh) upon reasonable request.

**Acknowledgments:** Thanks to Engr. Ziad Nayef Shatnawi of King Faisal University, Saudi Arabia for helping in preparation of some of the figures in the manuscript.

**Conflicts of Interest:** The authors would like to declare that there are no conflict of interest.

## Appendix A

**Table A1.** Sample of crash data used for modeling.

| Crash ID | Crash Date-Day | Crash Date-Month | Crash Date-Year | Crash Severity ID | Crash Type ID | Crash Reason ID | Number of Deaths | Number of Injuries Major |
|---|---|---|---|---|---|---|---|---|
| 1 | 9 | 11 | 1436 | 2 | 1 | 14 | | 1 |
| 2 | 6 | 6 | 1436 | 1 | 1 | 15 | 1 | 5 |
| 3 | 27 | 4 | 1436 | 1 | 9 | 14 | 1 | |
| 4 | 6 | 2 | 1436 | 2 | 1 | 4 | | 3 |
| 5 | 16 | 3 | 1437 | 1 | 1 | 15 | 4 | 46 |
| 6 | 6 | 2 | 1436 | 2 | 1 | 4 | | 2 |
| 7 | 2 | 1 | 1436 | 2 | 9 | 14 | | 2 |
| 8 | 15 | 2 | 1436 | 2 | 1 | 14 | | 1 |
| 9 | 16 | 2 | 1436 | 1 | 1 | 4 | 2 | 1 |
| 10 | 20 | 2 | 1436 | | 9 | 14 | | 1 |
| 11 | 23 | 1 | 1437 | 2 | 1 | 4 | | 1 |
| 12 | 8 | 5 | 1436 | 1 | 1 | 15 | 1 | 1 |
| 13 | 5 | 4 | 1436 | 1 | 1 | 4 | 1 | 2 |

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
