# Peer review of "Involvement of Road Users from the Productive Age Group in Traffic Crashes in Saudi Arabia: An Investigative Study Using Statistical and Machine Learning Techniques"

_applsci, doi:10.3390/app12136368_

Round 1

Reviewer 1 Report

This is an interesting paper. Here are some comments from the reviewer.

First, what are the key and novel contributions of this study? Both binary logistic regression model and CART model have been used in previous studies.

Second, some deep learning models may provide better prediction results.

Third, the prediction results are not very satisfactory, the authors are suggested to group the data into homogenous groups and then predict. The following study should be reviewed and considered: Vehicle Acceleration Prediction Based on Machine Learning Models and Driving Behavior Analysis. Appl. Sci. 2022, 12, 5259. https://doi.org/10.3390/app12105259

Fourth, the crash data should be described with more details. How many observations? What are the crash characteristics?

Fifth, it is interesting to see both models have similar prediction performances. Some state-of-the-art models should be included in the analysis.

Author Response

Dear Reviewer 1,

We would like to thank you and the reviewers for the comments on our manuscript titled “Involvement of Road Users from Productive Age Group in Traffic Crashes in Saudi Arabia: Investigative Study Using Statistical and Machine Learning Techniques”, ID: “applsci-1742266”. We have revised the manuscript carefully based on the constructive comments and suggestions given by you. A complete list of the revisions in the revised version of the manuscript is given in subsequent pages. We would like to thank you for your support on improving both the content and the structure of the manuscript.

Regards,

Md. Kamrul Islam *, Uneb Gazder *, Rocksana Akter, Md Arifuzzaman

Reviewer 2 Report

Review

of the article “Involvement of Road Users from Productive Age Group in

Traffic Crashes in Saudi Arabia: Investigative Study Using

Statistical and Machine Learning Techniques”

The topic of this article is relevant. The study is devoted to the investigation of the influence of various factors on the involvement of road users from productive age group in severe road accidents in Al-Ahsa city of Saudi Arabia using the developed predictive models. The article contains many related references, but without their comparative analysis. I recommend improving the manuscript according to the following remarks.

1. The article uses the term "Road traffic accidents (RTCs)", but the list of keywords includes "Road Traffic Accidents". The terms should be agreed.

2. Incorrect mathematical notation was used in the notation of formula (1). In addition, it would be appropriate to describe the relationship between the dependent variables of formulas (1) and (3): Pi and F (x).

3. In the explanation of formulas (2), (3) the vectors X and Y are given, and in the specified formulas their elements are written without indication of ranges of their indices.

4. It is necessary to substantiate the application of such a model of logistic regression, i.e., without regularization.

5. The numbering of formulas (1) - (4) is repeated in the text of the article.

6. Not all tree node captions in Figure 5 are clear. Appropriate clarifications are required.

7. The 2 figures are labeled "Figure 2".

8. It is advisable to remove "No. of "from the names of the second and third columns of Table 2.

9. What is the difference between the "annual rate for fatal crashes" and the "fatal crash frequency" (lines 237 and 238)?

10. There are grammatical errors and incorrect expressions, for example, the unclear abbreviation "RTAs" (lines 85 and 90), "Percision" instead of "Precision" (line 314).

In the end, the paper needs a minor revision according to the above comments.

Author Response

We would like to thank you for the comments on our manuscript titled “Involvement of Road Users from Productive Age Group in Traffic Crashes in Saudi Arabia: Investigative Study Using Statistical and Machine Learning Techniques”, ID: “applsci-1742266”. We have revised the manuscript carefully based on the constructive comments and suggestions given by you. A complete list of the revisions in the revised version of the manuscript is given in subsequent pages. We would like to thank you for your support on improving both the content and the structure of the manuscript.

Reviewer 3 Report

1. Firstly, the literature review hasn’t been concluded comprehensively, and the introduction also doesn’t state the scientific significance of this research. So, I think these creative points should be clearly concluded again.

2. Secondly, the sample of data resources should be presented in the article.

3. Then the reasons of selecting the binary logistic regression and classification and regression tree (CART) models don’t be illustrated in the manuscript. Why does this research compare them?

4. In the end, the conclusions should be summarized concisely, which are too long.

Author Response

We would like to thank you for the comments on our manuscript titled “Involvement of Road Users from the Productive Age Group in Traffic Crashes in Saudi Arabia: An Investigative Study Using Statistical and Machine Learning Techniques”, ID: “applsci-1742266”. We have revised the manuscript carefully based on the constructive comments and suggestions given by you. A complete list of the revisions in the revised version of the manuscript is given in subsequent pages. We would like to thank you for your support on improving both the content and the structure of the manuscript.

Round 2

Reviewer 1 Report

The authors are suggested to further revise the writing and correct the typos in the manuscript. For example,

Reference 49: Y. Zou, L. Ding, H. Zhang, T. Zhu, and L. Wu, “applied sciences Vehicle Acceleration Prediction Based on Machine Learning Models and Driving Behavior Analysis,” 2022 should be "Zou, Y., Ding, L., Zhang, H., Zhu, T., & Wu, L. (2022). Vehicle acceleration prediction based on machine learning models and driving behavior analysis. Applied Sciences, 12(10), 5259." 

Author Response

We would like to thank you and the reviewers for the comments on our manuscript titled “Involvement of Road Users from the Productive Age Group in Traffic Crashes in Saudi Arabia: An Investigative Study Using Statistical and Machine Learning Techniques”, ID: “applsci-1742266”. We have revised the manuscript carefully based on the constructive comments and suggestions given by you. A complete list of the revisions in the revised version of the manuscript is given in subsequent pages. We would like to thank you for your support on improving both the content and the structure of the manuscript.
